# Natural and Synthetic Compounds Against Colorectal Cancer: An Update of Preclinical Studies in Saudi Arabia

**DOI:** 10.3390/curroncol32100546

**Published:** 2025-09-29

**Authors:** Mansoor-Ali Vaali-Mohammed, Adhila Nazar, Mohamad Meeramaideen, Saleha Khan

**Affiliations:** 1Department of Zoology, Jamal Mohamed College (Autonomous), Affiliated to Bharathidasan University, Tiruchirappalli 620020, India; drmaideen84@gmail.com; 2Colorectal Cancer Research, Surgery Department, College of Medicine, King Saud University, Riyadh 11472, Saudi Arabia; 3Department of Pharmacy, Nehru College of Pharmacy, Affiliated to Kerala University of Health Sciences, Thrissur 680541, India; 4General Medicine Practice Program, Batterjee Medical College, Jeddah 21442, Saudi Arabia; salehashafi5@gmail.com

**Keywords:** colorectal cancer, Saudi Arabia, natural products, synthetic compounds, therapeutic strategies, nanotechnology, chemoresistance

## Abstract

Colorectal cancer is one of the most common and deadly cancers worldwide, and its cases are rising in Saudi Arabia, especially among younger people. Scientists are now exploring new ways to fight this disease using natural sources like plants and marine extracts, along with synthetic (lab-made) compounds. This article reviews recent studies from Saudi Arabia that tested these substances in lab and animal models to see how well they work against colorectal cancer. Some natural extracts, including black seed and Moringa, showed the ability to kill cancer cells and slow tumor growth. Synthetic drugs and specially designed nanoparticles also proved effective, particularly when combined with current treatments. These early findings are promising but have not yet been tested in people. More research is needed to confirm their safety and effectiveness in humans. By highlighting these discoveries, this review supports the search for better, more targeted cancer treatments. It also encourages further research in Saudi Arabia, using both natural resources and modern science aims to improve outcomes for patients with colorectal cancer.

## 1. Introduction

Colorectal cancer (CRC) remains one of the most prevalent and deadly malignancies worldwide, ranking among the top three causes of cancer-related mortality [1]. Globally, colorectal cancer accounts for approximately 1.9 million new cases annually, underscoring the disease’s significant burden on public health systems [2]. In Saudi Arabia, CRC has become the most commonly diagnosed cancer among men and the third most common among women. Notably, Saudi Arabia is experiencing a concerning rise in early-onset CRC, with increasing incidence reported among individuals under the age of 50 [2]. This trend reflects a shift in disease epidemiology and calls for region-specific prevention and treatment strategies.

The etiology of CRC involves a complex interplay of genetic, environmental, and lifestyle factors. Risk factors such as obesity, high-fat and low-fiber diets, physical inactivity, tobacco use, and genetic predispositions have all been implicated [3]. Conventional treatments—including surgery, chemotherapy, radiation therapy, and molecularly targeted agents have improved survival rates in some patient groups. However, treatment efficacy remains limited in advanced CRC due to challenges such as drug resistance, off-target toxicity, and tumor heterogeneity [4,5]. These limitations highlight the urgent need for new therapeutic approaches that are more precise, tolerable, and capable of overcoming resistance mechanisms.

Indicators of anti-cancer activity in CRC models typically include induction of apoptosis, cell cycle arrest, inhibition of angiogenesis, suppression of oncogenic signaling pathways, and modulation of oxidative stress responses. Clarifying these endpoints is essential to evaluate the therapeutic relevance of both natural and synthetic compounds.

In recent years, both natural and synthetic compounds have emerged as promising candidates for CRC therapy. Natural products derived from medicinal plants, marine organisms, and microorganisms exhibit anti-inflammatory, cytotoxic, and pro-apoptotic properties [6]. Key classes of compounds like alkaloids, flavonoids, terpenes, and polyphenols have shown anti-tumor effects through modulation of apoptosis, cell cycle arrest, and inhibition of oncogenic pathways such as NF-κB, EGFR, and COX-2 [7,8]. In parallel, the development of synthetic compounds, including small-molecule inhibitors and targeted therapies, has opened new avenues for CRC management [9]. Simultaneously, advances in medicinal chemistry have enabled the development of synthetic and semisynthetic molecules tailored to selectively target CRC-associated molecular pathways [10]. Moreover, novel drug delivery systems, particularly nanoparticle-based technologies, offer enhanced bioavailability and tumor-targeted delivery, further amplifying therapeutic outcomes [11,12].

Focusing this review on Saudi-based research is both timely and relevant. Saudi Arabia is experiencing a rising burden of colorectal cancer, particularly among younger individuals, which may reflect unique regional risk factors such as dietary shifts, genetic predispositions, and environmental exposures. Additionally, the country is actively expanding its biomedical research output under national strategies like Vision 2030, leading to an increasing number of preclinical oncology studies from local institutions [13]. By highlighting research conducted in Saudi Arabia, this review not only emphasizes the therapeutic potential of regionally available resources but also supports the advancement of localized, evidence-based cancer treatments. Understanding these region-specific findings may ultimately inform both national and global CRC management strategies.

Saudi Arabia’s rich biodiversity of medicinal plants and access to Red Sea marine ecosystems have facilitated the discovery of novel natural compounds. Concurrently, there is a growing body of Saudi-based research evaluating both natural and synthetic compounds for anti-CRC properties. Examples include *Nigella sativa*, *Moringa oleifera*, and marine-derived metabolites that induce apoptosis and cell cycle arrest and even overcome multidrug resistance in CRC cells [7,14].

For instance, the marine bacterium *Halomonas meridiana* has been identified as a producer of *L-glutaminase*, an enzyme capable of depriving cancer cells of glutamine and inducing apoptosis [15]. On the synthetic side, compounds such as C4, C16, and benzoxazole derivatives have demonstrated cytotoxic activity in CRC cell lines, suggesting potential for drug development [16]. These findings collectively highlight the depth and innovation of Saudi contributions to CRC research.

Additionally, accumulating evidence supports the role of phytochemical compounds from medicinal plants in activating anti-apoptotic and anti-tumor pathways, highlighting their potential not only in CRC therapy but also in chemoprevention strategies [17].

This review presents a comprehensive synthesis of preclinical research from Saudi Arabia investigating natural and synthetic compounds for colorectal cancer treatment. Emphasis is placed on their mechanisms of action, molecular targets (e.g., NF-κB, ROS, EGFR), nanocarrier innovations, and translational potential. Rather than presenting isolated findings, the review categorizes compounds by type and model, analyzes mechanistic convergence, and identifies key gaps in the current literature. By highlighting region-specific contributions and evaluating therapeutic promise, this review aims to inform future clinical validation and support the development of evidence-based CRC treatment strategies tailored to the Saudi context.

## 2. Methods

This review was conducted using a structured literature search and selection process. Electronic databases including PubMed, Scopus, Web of Science, and Google Scholar were searched to identify relevant studies. The search covered articles published between 2010 and 2025 to capture the most recent preclinical evidence.

A combination of keywords and MeSH terms was applied such as colorectal cancer, colon cancer, Saudi Arabia, natural products, synthetic compounds, nanoparticles, apoptosis, and anti-cancer activity. Boolean operators were used to refine search results, and the reference lists of retrieved studies were also screened to identify additional publications. The inclusion criteria considered studies conducted by researchers affiliated with Saudi-based institutions that investigated natural, synthetic, or nano-formulated compounds with anti-colorectal cancer activity in preclinical in vitro or in vivo settings. Only articles published in English in peer-reviewed journals were included. Exclusion criteria consisted of reviews, editorials, or conference abstracts without full data as well as studies not involving colorectal cancer models or lacking mechanistic or outcome-related data. Titles and abstracts were screened independently by the authors followed by full-text evaluation. Discrepancies in study selection were resolved through discussion and consensus. Data extracted from eligible studies included the type of compound, experimental model, molecular targets, mechanisms of action, and therapeutic outcomes.

The review synthesizes findings into thematic categories including natural products, synthetic compounds, and nanotechnology-enhanced formulations with emphasis on mechanistic pathways and translational potential.

## 3. Natural Products and Their Application

Natural products have emerged as potent therapeutic candidates in CRC treatment due to their broad spectrum of bioactive compounds with anti-inflammatory properties. This section discusses key findings from in vitro and in vivo studies conducted in the region. It is important to note that these compounds have not yet been tested in humans, and most of the current findings are preclinical and hypothesis-generating rather than clinically validated.

### 3.1. In Vitro Studies

#### 3.1.1. *Nigella sativa* (Black Seed) and Crude Saponin Extract (CSENS)

As previously introduced, *Nigella sativa* (black seed) is one of the most widely studied plant-derived agents in Saudi-based CRC research. In vitro studies using Crude saponin extract (CSENS) have demonstrated its anti-proliferative and from this plant exhibited anti-proliferative and apoptotic effects against HCT116 cells. These effects are primarily mediated through NF-Κb inhibition and Bax/Bcl-2 regulation [7], supporting its potential as an adjunct in CRC therapy.

#### 3.1.2. Olive Leaf Extract (*Olea europaea*)

Olive leaf extract, rich in chlorogenic acid, has been shown to inhibit the growth and migration of HT29 colorectal cancer cells. Experimental data indicate that the extract induces DNA fragmentation, causes S-phase cell cycle arrest, and elevates reactive oxygen species (ROS) production, supporting its potential as an anti-cancer agent [18].

#### 3.1.3. *Ferula hermonis* Root Extract (FHRH)

The root hexane extract of *Ferula hermonis* demonstrated dose-dependent cytotoxicity against (LoVo) colon cancer cells. Mechanistic studies revealed that FHRH triggers apoptosis through caspase 3/7 activation and modulates gene expression involved in cell death pathways [19].

The first group of natural agents—including *Nigella sativa*, olive leaf extract, and *Ferula hermonis*—shows considerable promise in inducing apoptosis and modulating inflammatory pathways such as NF-κB and COX-2. However, most studies evaluated these compounds using single CRC cell lines and short-term exposures, which limits conclusions about their comparative potency or long-term efficacy. Notably, *Nigella sativa* appears to be more extensively studied, yet lacks advanced mechanistic validation like siRNA knockdown or pathway inhibition. In contrast, *Ferula hermonis* demonstrates stronger anti-proliferative effects but without detailed dose–response profiling. This highlights a need for standardized protocols and broader mechanistic studies to draw more meaningful comparisons.

#### 3.1.4. Flavonoids in Colorectal Cancer Treatment

Flavonoids, a diverse group of polyphenolic compounds, are well-regarded for their anti-cancer potential due to their ability to promote apoptosis, inhibit proliferation, and modulate cellular pathways. Their low toxicity and favorable bioavailability make them candidates warranting further preclinical exploration in CRC models [8].

#### 3.1.5. Green Synthesized Silver Nanoparticles (AgNPs)

The application of green synthesis for the development of silver nanoparticles (AgNPs) using *Lasiurus scindicus* and *Panicum turgidum* seed extracts has been explored for its anticancer potential. Characterized through DLS and TEM techniques, this biosynthesized nanotechnology with phytochemicals marks a promising strategy for enhancing CRC treatment through improved delivery and reduced toxicity at the experimental level [20].

#### 3.1.6. *Selaginella repanda* Ethanolic Extract

Ethanolic crude extract of *Selaginella repanda* has demonstrated potent anticancer effects against HCT116 cells in a manner that is both dose- and time-dependent. Phytochemical analysis identified the presence of key bioactive compounds, including flavonoids, alkaloids, terpenoids, and phenolics, contributing to its observed efficacy. Its favorable pharmacokinetic profile and low toxicity suggest potential for further development as a CRC therapeutic [21].

#### 3.1.7. *Sansevieria trifasciata* Extract

The ethanolic extract of *Sansevieria trifasciata* leaves has been observed to selectively target HCT116 colorectal cancer cells, demonstrating higher cytotoxicity compared to normal colon epithelial cells. The significantly low IC_50_ values indicate its potential as a safe and effective candidate for CRC treatment [22].

#### 3.1.8. *Tetraclinis articulata* Essential Oil

Essential oil extracted from the trunk bark of *Tetraclinis articulata* displayed its moderate cytotoxic activity against SW620 colorectal carcinoma cells. The fractions with IC_50_ values below 30 μg/mL were particularly active, primarily due to the presence of oxygenated sesquiterpenes, such as caryophyllene oxide and carotol [23].

The next group of plant-derived compounds —flavonoids, *Selaginella repanda*, *Sansevieria trifasciata*, *Tetraclinis articulata*, and *Moringa oleifera*—exhibits diverse mechanisms, including ROS generation, caspase activation, and cell cycle arrest. While the green-synthesized AgNPs enhanced bioavailability and showed improved cytotoxicity, their safety and selectivity remain underexplored. *Moringa oleifera* showed the most mechanistic depth, influencing both intrinsic and extrinsic apoptotic pathways, though comparative efficacy against other agents was not directly assessed. Moreover, many of these studies do not report IC_50_ values or test across multiple CRC cell lines, making it difficult to assess relative potency or therapeutic index.

#### 3.1.9. *Moringa oleifera* Leaf and Bark Extracts

Extracts from the leaf and bark of *Moringa oleifera*, collected from Saudi Arabia, have demonstrated promising anticancer activity against HCT-8 colorectal cancer cells. Experimental findings revealed a reduction in cell viability, induction of apoptosis, and arrest in the G2/M phase of the cell cycle. Phytochemical analysis using GC-MS identified eugenol and D-allose as the primary active components contributing to its anti-cancer properties [14]. These cell cycle-related effects are illustrated in Figure 1.

#### 3.1.10. *Ziziphus nummularia* Ethanolic Extract

The ethanolic extract of *Ziziphus nummularia* was found to exert strong anti-cancer activity against HCT8 cells through mechanisms involving apoptosis and microtubule distribution. Luteolin-7-O-glucoside, identified as a key active compound, was shown to inhibit tubulin polymerization, causing M-phase arrest, positioning this plant as a potential source for novel anticancer agents, highlighting mechanistic novelty but remaining confined to in vitro testing [24].

#### 3.1.11. *Rhazya stricta* Alkaloid Extract (CAERS)

Alkaloid extract from *Rhazya stricta* demonstrated significant inhibition of cell proliferation and induced apoptosis in HCT116 colorectal cancer cells. Mechanistic analysis indicated that CAERS downregulated NF-κB, AP-1, and ERK MAPK pathways while enhancing the expression of pro-apoptotic markers p53, p21, Bax, and caspases, supporting its role as a chemotherapeutic agent [25].

In addition to specific plant extracts, alkaloids, a key class of phytoconstituents, have been intensively studied for their ability to promote ROS-mediated apoptosis in colon cancer cells. These chemicals increase intracellular ROS levels, alter redox equilibrium, and initiate apoptotic pathways that target factors like IGF-1, making them a viable therapeutic strategy with low toxicity to normal cells in pre-clinical models [26].

Among all tested compounds, most share common pathways of action—namely, caspase activation, ROS elevation, and suppression of anti-apoptotic signals such as Bcl-2. However, the heterogeneity in experimental design, cell lines used, and reporting metrics poses a major barrier to inter-study comparisons. Additionally, none of these in vitro studies incorporates co-treatment with standard chemotherapies to evaluate synergistic or antagonistic interactions, which are crucial for potential clinical application.

As summarized in Table 1, most plant-derived compounds exert their cytotoxic effects via induction of apoptosis and oxidative stress, frequently involving the NF-κB pathway, ROS generation, and caspase activation. These shared mechanistic features suggest a common underlying mode of action across structurally diverse phytochemicals, which may allow for synergistic combinations or shared delivery strategies in future research.

The collective mechanisms of action through which these compounds exert anticancer effects in CRC are illustrated in Figure 2.

This diagram illustrates how bioactive compounds such as Nigella sativa, *Moringa oleifera*, IMF-8, and C4–G4 interact with key molecular targets in CRC cells. Arrows represent upregulation (↑) or downregulation (↓) of specific proteins and pathways: Bax, Caspase-3, NF-κB, Bcl-2, COX-2, EGFR, and ROS. For example, *Nigella sativa* increases Bax and Caspase-3 while reducing NF-κB and Bcl-2; *Moringa oleifera* induces ROS and causes G2/M phase arrest in HCT-8 cells; IMF-8 enhances ROS and suppresses COX-2 in vivo; and C4–G4, dual inhibitors of EGFR and COX-2, target HT29 cells. These molecular effects collectively contribute to apoptosis, inhibition of cell proliferation, and tumor suppression in CRC. Hence, across all tested natural products, overlapping mechanistic targets include NF-κB inhibition, ROS generation, caspase activation, and suppression of Bcl-2. *Nigella sativa* and *Rhazya stricta* share NF-κB suppression, but Rhazya shows broader modulation of ERK MAPK and p53. *Moringa oleifera* and *Ziziphus nummularia* stand out for mechanistic novelty, inducing phase-specific cell cycle arrest (G2/M and M-phase). *Ferula hermonis* shows strong anti-proliferative effects but lacks robust dose–response profiling. Olive leaf extract provides clear ROS-linked S-phase arrest but is underexplored across different CRC lines. AgNPs improve delivery and cytotoxicity but remain poorly evaluated for safety and selectivity. The main gaps across studies include reliance on single CRC cell lines, lack of standardized dose–response assays, and absence of co-treatment studies with standard chemotherapies. These findings are valuable but remain exploratory and preclinical, requiring validation in advanced models before translational conclusions can be drawn.

### 3.2. In Vivo Studies

While extensive in vitro studies have demonstrated the potential of natural products against colorectal cancer (CRC), fewer in vivo studies have been conducted, particularly within Saudi Arabia. The following subsections detail the promising findings from animal models, emphasizing the anti-tumor capabilities of various natural extracts. All four studies summarized below were conducted by research groups based in Saudi Arabia or utilized compounds collected from Saudi flora.

#### 3.2.1. *Ferula hermonis* Root Extract (FHRH)

Preclinical investigations have explored the anti-cancer properties of *Ferula hermonis* root extract in rodent models. While tested in a DMBA-induced mammary tumor model, the extract’s tumor-suppressing activity provides preliminary rationale for future CRC-specific applications [19].

#### 3.2.2. *Ferula assa-foetida* OGR Extract

The oleo-gum-resin (OGR) extract of *Ferula assa-foetida* has been evaluated for its anti-tumor properties in a xenograft mouse model using HT-29 colorectal cancer cells. Treatment with this extract resulted in a marked reduction in tumor volume, underscoring its potential as a therapeutic agent for CRC [27].

#### 3.2.3. *Arthrocnemum machrostachyum* Methanolic Extract (AME)

Methanolic extract of *Arthrocnemum machrostachyum* demonstrated strong anti-tumor activity in an Ehrlich solid tumor mouse model. Administration of AME significantly decreased tumor size, induced apoptosis, and regulated key apoptotic markers, including p53, Bax, and caspase-3. Additionally, anti-inflammatory effects were observed through the suppression of TNFα expression, highlighting its potential as an anticancer adjuvant [28].

#### 3.2.4. Curcumin Supplementation in AOM-DSS Mouse Model

Curcumin, a bioactive compound derived from *Curcuma longa*, has demonstrated potent anti-inflammatory and tumor-suppressive activity in an AOM-DSS-induced colorectal cancer mouse model fed a high-protein diet. The treatment significantly reduced tumor multiplicity, mitigated colonic inflammation, inhibited colonocyte proliferation, and decreased toxic metabolite production. While traditionally considered a chemopreventive agent, these findings also underscore its therapeutic potential in modulating the tumor microenvironment and halting CRC progression after initiation. The ability of curcumin to influence pro-inflammatory and proliferative pathways supports its candidacy for adjuvant therapy in CRC, especially in populations exposed to high-risk dietary patterns [29].

The in vivo studies—featuring *Ferula hermonis*, *Ferula assa-foetida*, *Arthrocnemum macrostachyum*, and curcumin—highlight the anticancer potential of several natural products through mechanisms such as apoptosis induction, modulation of pro-apoptotic genes, and suppression of tumor-promoting inflammation. Despite these promising findings, certain limitations must be acknowledged. The animal models used, including DMBA-induced tumors, xenograft mice, and Ehrlich solid tumor models, while useful, do not fully replicate human CRC pathophysiology, especially regarding immune and microenvironmental interactions. Notably, only the curcumin study used an established CRC-specific model (AOM-DSS), and none were conducted using CRC models tailored to the genetic or environmental characteristics of the Saudi population. Additionally, variations in animal strain, dosing strategies, and outcome measures limit direct comparison of therapeutic efficacy across studies. Key pharmacokinetic data and toxicity profiles were not consistently reported, which are essential for evaluating translational potential.

To enhance clinical relevance, future studies should utilize orthotopic or genetically engineered CRC models. Standardized evaluation protocols and inclusion of region-specific CRC factors will further strengthen translational potential.

Across the natural products reviewed, apoptosis induction, ROS generation, and NF-κB inhibition emerged as common mechanistic pathways. These overlapping mechanisms suggest that, despite structural diversity, these compounds may exert cytotoxicity via shared signaling disruptions. The convergence on intrinsic apoptosis highlights potential for synergistic combinations, but the lack of standardization in model selection and concentration ranges hinders comparative evaluation.

As summarized in Table 2, apoptosis induction and inflammation suppression are the predominant mechanisms among in vivo-tested natural compounds. However, mechanistic validation remains limited, with only curcumin evaluated in the CRC-specific model. This underscores a need for more rigorous in vivo mechanistic studies and model standardization.

These findings are visually summarized in Figure 3, which highlights the anti-tumor effects of FHRH, OGR, AME, and curcumin extracts across different in vivo colorectal cancer models. Common pathways included apoptosis induction and inflammatory modulation, although only curcumin was tested in a CRC-specific model.

## 4. Synthetic Compounds and Their Application

Recent advancements in colorectal cancer (CRC) research have highlighted the therapeutic potential of synthetic and semi-synthetic compounds. These engineered molecules are specifically designed to target critical signaling pathways involved in CRC development, metastasis, and treatment resistance. Notably, synthetic compounds have demonstrated the capability to inhibit pathways such as EGFR, COX-2, and β-catenin while also enhancing the efficacy of traditional chemotherapies.

### 4.1. In Vitro Studies

#### 4.1.1. Phenolic Acid Derivatives (C1-C4, P1-P4, G1-G4)

A series of semi-synthetic derivatives originating from phenolic acids in *Amaranthus spinosus* have been developed to act as dual inhibitors of EGFR and COX-2. Among the synthesized compounds, C4 and G4 exhibited significant cytotoxic effects against HT-29 colorectal cancer cells, achieving IC_50_ values of 0.9 μM for EGFR and 0.5 μM for COX-2, respectively. These findings highlight their potential as dual-target inhibitors in CRC therapy [30]. C4 and G4 are semi-synthetic hybrid molecules combining synthetic pharmacophores (benzoxazole, benzimidazole, and 4-hydroxyacetophenone) with natural flavonoid moieties such as quercetin and genistein. These hybrids function as dual inhibitors of epidermal growth factor receptor (EGFR) and cyclooxygenase-2 (COX-2), two key targets implicated in colorectal cancer proliferation and inflammation. Their respective IC_50_ values against HT-29 colorectal cancer cells were 0.9 μM (C4 for EGFR) and 0.5 μM (G4 for COX-2), indicating strong potency and selectivity. The fusion of synthetic cores and natural scaffolds improves binding affinity and may reduce resistance mechanisms in CRC models.

The structural rationale for these hybrid inhibitors, combining synthetic scaffolds and natural flavonoid cores, is illustrated in Figure 4.

#### 4.1.2. Silver Nanoparticles from Chamomile Flower Extract (SN-CHM)

Biogenic synthesis of silver nanoparticles (SN-CHM) using chamomile flower extract has emerged as a novel anticancer approach. These nanoparticles exhibited stable morphology (~115 nm), negative surface charge, and antioxidant properties. In vitro studies demonstrated significant cytotoxicity against SW620 and HT-29 colorectal cancer cells, suggesting their potential as eco-friendly and effective anticancer agents [31].

#### 4.1.3. Withaferin-A and 5-Fluorouracil Combination

Combining withaferin-A (WA), a natural steroidal lactone, with 5-fluorouracil (5-FU) demonstrated enhanced anti-cancer efficacy in colorectal cancer cells. This combination effectively induced apoptosis through ER stress-mediated mechanisms, downregulated β-catenin signaling, and triggered G2/M cell cycle arrest. These results indicate a synergistic interaction, potentially overcoming resistance barriers in CRC treatment [32].

#### 4.1.4. Myricetin-Conjugated Silver Nanoparticles

The conjugation of the natural flavonoid myricetin with silver nanoparticles has been explored for its enhanced cytotoxic effects against CRC cells. Characterization confirmed successful nanoparticle synthesis. Treated CRC cells exhibited notable morphological changes and increased apoptosis, suggesting enhanced cytotoxic efficacy. These findings suggest that nanotechnology-based delivery of myricetin may improve its therapeutic efficacy [33].

#### 4.1.5. *Adansonia digitata* Polar Extract

The polar extract of *Adansonia digitata* (baobab) fibers has been reported to possess significant anti-proliferative effects against HCT116 colorectal cancer and MCF-7 breast cancer cells. Mechanistic studies revealed its capacity to modulate gene expression linked to tumor growth suppression, positioning it as a novel phytochemical lead for further exploration in CRC therapy [34].

#### 4.1.6. Green-Synthesized Cobalt Oxide Nanoparticles

Phytochemical synthesis of cobalt oxide nanoparticles using *Psidium guajava* leaf extract demonstrated potent anti-cancer effects against HCT116 colorectal and MCF-7 breast cancer cells, with minimal toxicity to normal cells. This suggests that green-synthesized nanoparticles could be effective, sustainable alternatives in CRC treatment [35].

#### 4.1.7. Oxazole Derivatives

A novel series of oxazole derivatives has shown anti-proliferative effects against HCT116 colorectal cancer cells, with compound 14 emerging as the most effective (IC_50_ = 71.8 μM). These findings indicate the therapeutic potential of oxazole analogs as novel anti-cancer agents [36].

#### 4.1.8. Camptothecin-Encapsulated Nanocarriers (CEFs)

To enhance its bioavailability and therapeutic effectiveness, camptothecin *(CPT)* has been encapsulated within a cyclodextrin-EDTA-Fe_3_O_4_ (CEF) composite nanoparticle. This innovative formulation has demonstrated dose-dependent cytotoxicity against HT29 colorectal cancer cells, primarily through caspase-3-mediated apoptosis and G1 phase cell cycle arrest. The application of magnetic nanocarriers in this manner not only improves solubility but also enables targeted delivery, reducing systemic toxicity and enhancing anti-cancer efficacy [37].

#### 4.1.9. Sipholenol A Derivatives for MDR Reversal

The emergence of multidrug resistance (MDR) in colorectal cancer remains a significant therapeutic challenge. Semi-synthetic derivatives of Sipholenol *A*, including 4-O-acetate and 4-O-isonicotinate, have shown the ability to overcome P-glycoprotein-mediated MDR. These derivatives enhance intracellular drug accumulation by stimulating ATPase activity and displaying strong interactions with P-glycoprotein (P-gp), a key efflux transporter involved in chemoresistance. This highlights their potential as effective agents for reversing MDR in CRC and other cancers [38].

Among the synthetic compounds examined in vitro, the semi-synthetic derivatives C4 and G4 displayed the most effective anticancer effects, acting as dual inhibitors of EGFR and COX-2 with broad cytotoxic action across CRC and other solid tumor cell lines. Their multitargeted mechanism makes them intriguing candidates for preclinical development. In contrast, the oxazole analog compound 14 displayed somewhat lesser activity, with an IC_50_ of 71.8 µM in HCT116 cells, significantly higher than conventional cut-offs for drug-like potency. However, robust CDK8 binding shows potential as a lead structure. Other treatments, such as Withaferin-A combinations and camptothecin nanocomposites, provided promising mechanistic insights (e.g., ER stress, β-catenin suppression, caspase-3 activation) but require further validation in dose–response studies and in vivo models. At present, only C4 and G4 exhibit the pharmacological profile and molecular targeting selectivity needed to justify progression to preclinical animal testing. However, none of the reviewed compounds have undergone pharmacokinetic profiling, formulation testing, or bioavailability assessments—critical steps for clinical translation.

Synthetic agents more frequently targeted specific signalling axes such as EGFR, COX-2, PI3K/Akt, and CDK pathways. These defined targets offer greater translational feasibility and optimization potential, yet the absence of head-to-head studies and long-term safety data makes it difficult to rank their clinical readiness. Further investigation is needed to compare synthetic versus natural agents under unified experimental conditions.

As summarized in Table 3, synthetic compounds generally exhibit more targeted pathway inhibition (e.g., dual EGFR and COX-2 suppression by C4 and G4) and lower IC_50_ values compared to natural compounds. These agents also demonstrate better-defined structures and molecular compatibility, indicating stronger potential for preclinical optimization.

### 4.2. In Vivo Studies

To evaluate the translational potential of synthetic compounds in colorectal cancer (CRC) therapy, several in vivo models have been employed to investigate their efficacy, mechanisms of action, and impact on tumor progression. The following studies highlight key findings that underscore their promise in CRC treatment.

While not all synthetic agents were developed within Saudi institutions, several studies included here originate from Saudi-based collaborations or used regionally relevant CRC models, supporting the review’s localized research scope.

#### 4.2.1. [V4Q5]dDAVP in Combination with 5-FU

[V4Q5]dDAVP, a synthetic analog of vasopressin targeting AVPR2 receptors, has been studied for its ability to enhance the anticancer effects of low-dose 5-fluorouracil (5-FU) in colorectal cancer models. In both CT-26 murine and COLO-205 human xenografts, the combination therapy significantly inhibited tumor growth, promoted apoptosis, and reduced lung metastasis. Mechanistically, this enhancement was linked to increased expression of p21 and p53, suggesting that [V4Q5]dDAVP could be an effective co-adjuvant in CRC therapy [39].

#### 4.2.2. Zotarolimus as a Co-Adjuvant Therapy

Zotarolimus, a semi-synthetic inhibitor of the mammalian target of rapamycin (mTOR), demonstrated marked anti-tumor effects in HCT-116 colorectal cancer xenograft models. When administered alone or in combination with 5-FU, it led to significant reductions in tumor volume and enhanced apoptotic signaling through cleaved caspase-3 and ERK pathways. Additionally, it downregulated key inflammatory and metastasis-associated proteins, highlighting its potential as a co-adjuvant in CRC management [40].

#### 4.2.3. IMF-8 (Iminoflavone Derivative)

IMF-8, a semi-synthetic derivative of iminoflavone, has been evaluated for its chemopreventive properties in a DMH-induced colorectal cancer rat model. Its administration resulted in a reduction in aberrant crypt foci, a decrease in polyp formation, and the modulation of oxidative stress markers such as catalase and GSH. Furthermore, it downregulated pro-inflammatory cytokines (TNF-α, IL-6), indicating its protective role in CRC prevention [41].

#### 4.2.4. WNT974 and Artesunate (ART) Combination Therapy

The combination of WNT974, a porcupine inhibitor, with artesunate (ART), a semi-synthetic artemisinin derivative, demonstrated significant anti-cancer activity in CRC xenograft models. This therapeutic pairing promoted the degradation of KRAS via the ubiquitin-proteasome pathway, upregulated E3 ligases (ANAPC2, β-TrCP), and inhibited PI3K/Akt/mTOR signaling. These findings position the dual therapy as a promising strategy for targeting KRAS-driven colorectal cancers [42].

#### 4.2.5. Potassium Koetjapate (KKA)

Potassium koetjapate (KKA), a semi-synthetic derivative of koetjapic acid, showed substantial anti-tumor activity in HCT116 colorectal cancer models. KKA was found to suppress tumor progression by downregulating anti-apoptotic markers (HSP60, Bcl-2, IGF-1) and upregulating apoptotic pathways through caspase activation and TRAIL receptors. Additionally, it inhibited key signaling pathways, including MAPK, Notch, and Wnt, indicating its multi-targeted mechanism in CRC treatment [43].

#### 4.2.6. 20(S)-Protopanaxadiol (PPD)

20(S)-Protopanaxadiol (PPD), a metabolite of ginsenosides, has demonstrated robust anti-cancer effects in HCT116 xenograft models. Treatment with PPD significantly reduced tumor volume and cell proliferation by inhibiting NF-κB, JNK, and MAPK/ERK pathways. Further mechanistic studies revealed its ability to downregulate PITPNA and upregulate AKAP8L, highlighting its potential as a multi-targeted therapeutic agent in CRC [44].

Among the synthetic agents evaluated in vivo, Zotarolimus and 20(S)-Protopanaxadiol (PPD) exhibit the strongest therapeutic potential based on their ability to suppress key oncogenic pathways —including EGFR, VEGF, NF-κB, and MAPK/ERK—and significantly reduce tumor burden in CRC xenograft models. Zotarolimus also showed increased efficacy when coupled with 5-FU, indicating its appropriateness for combinatorial regimens. PPD regulated a greater number of targets, including PITPNA and AKAP8L, but its data are limited due to a lack of current validation and pharmacokinetic knowledge. IMF-8 and potassium koetjapate (KKA) exhibited pathway-specific cytotoxicity, mainly through oxidative stress modulation and TRAILR-mediated apoptosis but were not tested for dose tolerance or systemic toxicity. The WNT974-artesunate combination efficiently targeted the PI3K/Akt/mTOR axis and promoted KRAS degradation, which is desirable in resistant CRC cases, albeit data on long-term tumor suppression and survival are limited. Similarly, the [V4Q5]dDAVP-5FU combination increased overall survival and metastasis inhibition in mouse models; however, human relevance is questionable because of changes in vasopressin receptor biology.

All trials lack comparative data on pharmacokinetics, bioavailability, and formulation compatibility, making it impossible to determine clinical readiness. To determine clinical readiness, preclinical studies should incorporate orthotopic CRC models along with toxicity and pharmacokinetic profiling. These steps are essential before human testing.

A graphical representation of these key in vivo findings is provided in Figure 5, illustrating compound-specific tumor responses and molecular mechanisms.

As summarized in Table 4, most in vivo-tested synthetic agents act through well-characterized apoptotic and signaling pathways, including NF-κB, PI3K/Akt/mTOR, and VEGF inhibition. Several studies also include combination therapies with 5-FU, highlighting a translational trend toward synergistic regimens. Yet, the absence of long-term toxicity and pharmacokinetic data limits their clinical readiness.

## 5. Translational Relevance and Potential Clinical Applications

The clinical management of colorectal cancer (CRC) remains a significant challenge due to its high recurrence rates and resistance to conventional chemotherapy. In recent years, growing attention has been directed toward natural and synthetic compounds that exhibit promising anticancer activity in preclinical studies to explore their translational potential. Although not yet approved for clinical use, many of these agents have shown mechanisms of action that justify further investigation toward future therapeutic development. This section highlights compounds prioritized in Saudi-based research and assesses their potential for clinical development, despite being in early investigational phases.

### 5.1. Natural Products: Toward Clinical Translation

Natural products have historically played a critical role in drug discovery, especially in oncology. In the Saudi Arabian research context, various plant-derived compounds such as *Nigella sativa*, *Moringa oleifera*, *Ferula hermonis*, *Curcuma longa*, and *Rhazya stricta* have shown preclinical promise in CRC models. These agents modulate key signaling pathways, including NF-κB, AP-1, and caspase cascades, supporting their potential for clinical development [7,14]. Notably, *Curcuma longa*, traditionally recognized for its chemopreventive properties, has also demonstrated therapeutic efficacy in established CRC models, particularly through suppression of inflammation, reduction in colonocyte proliferation, and alteration of tumor microenvironment factors [29].

Additionally, marine-derived compounds, like L-glutaminase from *Halomonas meridiana*, have demonstrated nutrient-deprivation mechanisms targeting CRC cells [15]. Furthermore, silver nanoparticles synthesized from *Lasiurus scindicus* and *Panicum turgidum* are being explored as delivery platforms to enhance the cytotoxicity and selectivity of phytochemicals [20].

While these findings are encouraging, it is important to note that clinical trials are lacking, and most of these compounds remain in the preclinical phase. Their successful translation will require pharmacokinetic studies, toxicity profiling, and standardized formulations.

Recent advances in nanotechnology offer solutions to these limitations. As illustrated in Figure 6, PEGylated nanoparticles can facilitate the targeted delivery of natural and synthetic agents, such as myricetin, camptothecin, and IMF-8, by enhancing drug solubility, improving tumor selectivity, and minimizing systemic toxicity.

### 5.2. Synthetic Compounds and Preclinical Promise

Synthetic and semi-synthetic compounds have emerged as powerful tools to target molecular pathways relevant to CRC, such as EGFR, COX-2, and β-catenin. For example, C4 and G4 (dual inhibitors of EGFR and COX-2) and oxazole derivatives have shown promising in vitro results, with potential to overcome drug resistance and enhance efficacy of traditional therapies [30]. The combination of Withaferin-A and 5-FU overcame chemoresistance in vitro by inducing ER stress-mediated apoptosis. In parallel, oxazole derivatives effectively targeted CDK8 in HCT116 cells, highlighting their potential as promising therapeutic leads in CRC treatment [32,36].

In vivo models have demonstrated the potential of agents such as Zotarolimus, potassium koetjapate (KKA), and 20(S)-protopanaxadiol (PPD) in reducing tumor volumes and modulating pro-apoptotic pathways [40,43,44]. Additionally, the combination of WNT974 and artesunate (ART) has shown capability in vivo activity by degrading KRAS protein, one of the most challenging oncogenic drivers in colorectal cancer treatment in animals [42]. Although these findings are limited to preclinical models, they provide a compelling foundation for advancing these compounds toward clinical evaluation. The contribution of epithelial–mesenchymal transition (EMT) to colorectal cancer progression and its modulation by bioactive compounds [45] is illustrated in Figure 7.

### 5.3. Bridging Preclinical Research and Future Clinical Use

The ultimate goal of these investigations is to facilitate the translation of benchside findings to bedside applications. However, the current status of these compounds is exploratory and hypothesis-generating, not yet validated through human clinical trials.

To move forward, comprehensive pharmacological studies, safety evaluations, and early-phase clinical trials are essential. Saudi Arabia’s expanding biomedical research infrastructure, bolstered by national initiatives such as Vision 2030, offers a strategic platform.

Moreover, nanotechnology-based delivery systems, biomarker-guided selection (e.g., KRAS, EGFR, COX-2), and personalized medicine approaches hold promise in enhancing the clinical relevance of these natural and synthetic bioactives. Strategic collaborations between research institutions and industry will be key to accelerating this transition.

While both natural and synthetic compounds show promise against colorectal cancer, their pathways toward clinical translation differ significantly. Natural products, particularly crude extracts or bioactive fractions, face substantial regulatory and formulation challenges due to batch variability, poor bioavailability, and undefined pharmacokinetics. Most plant-based treatments lack standardization, stability, and dose consistency, which limits their clinical suitability. In contrast, synthetic compounds such as Zotarolimus, IMF-8, and PPD exhibit defined chemical structures, target selectivity, and dose control, making them better suited to formulation optimization and regulatory pathways. However, synthetic agents also require extensive safety profiling and often involve high development costs. Nanotechnology-based delivery systems offer a bridge for both categories by improving solubility, targeting precision, and therapeutic index. Ultimately, synthetic derivatives may progress more readily into early-phase trials, while natural compounds require further optimization and validation to meet clinical standards.

While these innovations offer exciting therapeutic avenues, several limitations must still be addressed to enable real-world application, as discussed in the following section.

## 6. Global Perspectives on Natural and Synthetic Anti-CRC Compounds

Global findings highlight the role of natural products as a cornerstone in CRC therapeutics. Multiple studies demonstrate that plant-derived polyphenols such as curcumin, resveratrol, and quercetin, marine-derived metabolites including peptides, alkaloids, and carotenoids, as well as synthetic agents like 5-fluorouracil, oxaliplatin, and irinotecan, act through overlapping mechanisms such as apoptosis induction, NF-κB and Wnt/β-catenin inhibition, and epigenetic modulation [46,47,48,49,50]. Importantly, some marine-derived compounds have advanced to clinical evaluation or approval, particularly as antibody–drug conjugates [49,51,52,53]. Saudi-based studies similarly highlight promising natural candidates such as Nigella sativa, *Moringa oleifera*, *Ferula hermonis*, *Curcuma longa*, and *Rhazya stricta*, with mechanistic profiles that parallel these global trends by targeting inflammatory pathways, caspase cascades, and tumor microenvironment factors [54]. Likewise, marine-derived compounds such as L-glutaminase from *Halomonas meridiana* reflect the global interest in nutrient-deprivation strategies for CRC cells [55]. Innovations such as silver nanoparticles synthesized from *Lasiurus scindicus* and *Panicum turgidum* to enhance phytochemical delivery also align with international efforts in nanomedicine-based CRC therapy [20]. Moreover, combination strategies pairing natural agents with synthetic drugs are increasingly recognized worldwide for enhancing efficacy and reducing toxicity [47,52,53]. Together, these findings, while mostly preclinical, suggest that Saudi contributions are conceptually consistent with global discoveries and have potential translational relevance. However, clinical trials remain critical to validate their therapeutic integration, standardize dosing, and characterize pharmacokinetics and toxicity.

## 7. Limitations, Future Perspectives, and Conclusion

This review focuses on the substantial advances made in the study of natural compounds and synthetic analogs as prospective therapeutic methods for colorectal cancer (CRC). However, various constraints continue to impede their transition from preclinical research to clinical application. Many of the substances examined, including Nigella sativa, *Moringa oleifera*, and marine-derived metabolites, have low bioavailability, poor solubility, and metabolic instability. These pharmacokinetic deficiencies restrict their systemic efficacy and pose problems to routine dosing, especially in human models.

Furthermore, while in vitro and in vivo models provide strong early evidence, the majority of the included studies are limited to laboratory settings, with little data from clinical trials. Agents such as Withaferin-A, Zotarolimus, and potassium koetjapate (KKA), while efficacious in animal models, have not been validated in human subjects, raising concerns about interspecies variability, safety, and long-term toxicity. Furthermore, discrepancies in extraction procedures, characterization techniques, and dosing protocols between research, particularly those done in Saudi Arabia, limit repeatability and preclude the development of consistent treatment guidelines.

Another unexplored area is the possibility of pharmacokinetic and pharmacodynamic interactions between these drugs and current chemotherapy regimens. To be clinically useful, therapies must investigate how these novel drugs act in combination, particularly in terms of toxicity, resistance mechanisms, and therapeutic synergy.

While the reviewed studies generally support the anticancer potential of natural and synthetic agents, inconsistencies across experimental designs, endpoints, and reporting standards reduce the reliability of inter-study comparisons. Some compounds demonstrated significant effects in vitro but lacked follow-up in vivo validation, while others were tested in animal models without mechanistic confirmation. Furthermore, variability in extraction methods, solvent systems, dosage ranges, and treatment durations—particularly across studies conducted in Saudi Arabia—contributes to conflicting findings. The absence of a head-to-head comparison, standardized protocols, and unified outcome metrics presents a major challenge in identifying the most promising candidates. Addressing these discrepancies through a harmonized research design and multi-center validation is essential to advance the field beyond descriptive observations.

To address these shortcomings, future research should focus on well-structured clinical studies that assess efficacy, safety, and tolerability in a variety of patient populations. Pharmacokinetic profiling and toxicity assessment must be part of such trials. The use of nanotechnology-based delivery vehicles, including liposomes, PEGylated nanoparticles, and polymeric carriers, could significantly improve the stability, solubility, and tumor-targeting capabilities of these medications. Studies in Saudi Arabia have already shown promise in this field, utilizing green-synthesized silver and cobalt oxide nanoparticles coupled with phytochemicals [31,35].

Immunotherapy, particularly through CRC-specific vaccines, immune checkpoint inhibitors, and CAR-T cell-based strategies, has shown significant potential in preclinical and early clinical studies to improve survival and reduce resistance in colorectal cancer patients [20,56].

Furthermore, combinatorial therapy techniques, such as combining synthetic drugs (e.g., IMF-8, 20(S)-protopanaxadiol) with traditional chemotherapeutics, have the potential to overcome multidrug resistance and improve treatment response [41,44]. The use of personalized medicine approaches driven by molecular biomarkers (KRAS, EGFR, COX-2) is another critical step toward enhancing patient selection, decreasing side effects, and optimizing therapeutic outcomes.

Saudi Arabia’s unique biodiversity and developing research infrastructure make it well-positioned to lead regional cancer research. National investment efforts, such as Vision 2030, have led to increased research output in the biomedical field. To continue this success, it is critical to foster collaboration among universities, pharmaceutical companies, and clinical facilities. The creation of cancer registries, pharmacogenomic databases, and biobanks would allow large-scale, population-based investigations suited to the Saudi population and similar Middle Eastern cohorts. To our knowledge, this is the first review to comprehensively synthesize natural and synthetic anti-CRC compounds from Saudi-based preclinical research, with a focus on nanocarrier development and translational feasibility.

International collaboration will also be important in developing CRC medication development. Unified clinical procedures, joint trials, and data sharing among regional and worldwide partners can help speed up the approval and adoption of these innovative drugs. Furthermore, post-marketing surveillance and long-term safety monitoring should be included in all stages of translational research to ensure that therapeutic advancements are both effective and sustainable.

Finally, combining natural products, marine bioactives, synthetic chemicals, and nanotechnology-based delivery systems is a viable technique for improving colorectal cancer treatment. These improvements, when led by predictive biomarkers and supported by thorough clinical research, have the potential to completely transform CRC management. To close the gap between laboratory findings and real-world clinical impact, ongoing investment in translational research, collaborative infrastructure, and patient-centered techniques is required.

## Figures and Tables

**Figure 1 curroncol-32-00546-f001:**
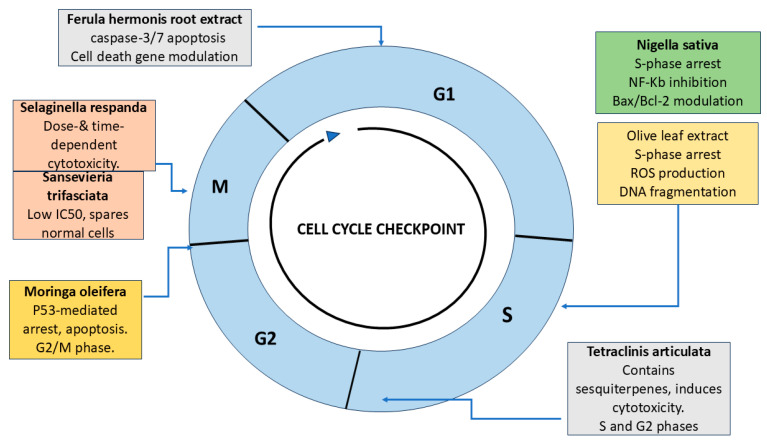
Cell cycle schematic showing the phases (G_1_, S, G_2_, and M) and key natural compounds investigated in Saudi-based studies for colorectal cancer (CRC) therapy. *Ferula hermonis* targets the G_1_ phase via caspase-3/7 activation and modulation of apoptotic gene expression. *Nigella sativa* and Olive leaf extract induce S-phase arrest through NF-κB inhibition, Bax/Bcl-2 modulation, ROS elevation, and DNA fragmentation. *Tetraclinis articulata* demonstrates cytotoxic activity in both S and G_2_ phases due to oxygenated sesquiterpenes. *Moringa oleifera* causes p53-mediated G_2_/M arrest. *Selaginella repanda* and *Sansevieria trifasciata* exert M-phase-specific effects, showing dose- and time-dependent cytotoxicity and selective action on CRC cells, respectively. Together, these agents modulate checkpoints and apoptotic pathways, supporting their relevance in CRC treatment strategies (created by the authors using PowerPoint).

**Figure 2 curroncol-32-00546-f002:**
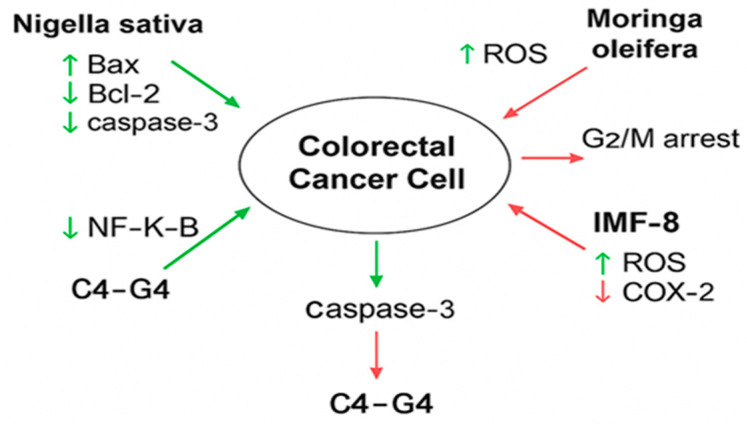
Mechanistic pathways modulated by selected natural and synthetic compounds in colorectal cancer (CRC) cell lines. Illustration prepared by the authors.

**Figure 3 curroncol-32-00546-f003:**
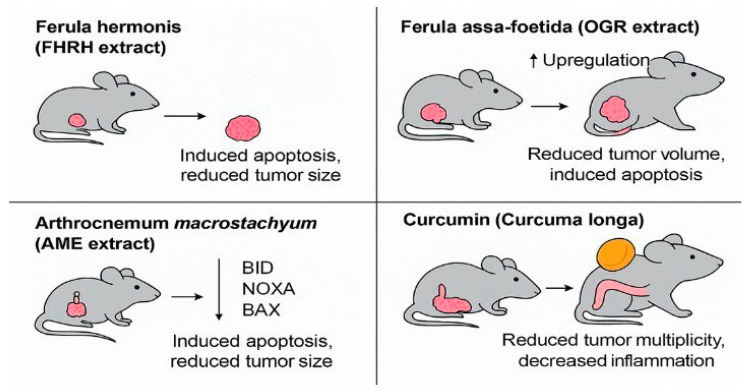
Mechanistic summary of in vivo anti-tumor effects of selected natural products in colorectal cancer models. The figure illustrates representative outcomes of preclinical animal studies evaluating Saudi-sourced or Saudi-investigated compounds. *Ferula hermonis* (FHRH extract) induced apoptosis and reduced tumor size in a DMBA-induced rodent model. *Ferula assa-foetida* (OGR extract) demonstrated volume reduction and pro-apoptotic gene upregulation in HT-29 xenografts. Arthrocnemum macrostachyum (AME extract) activated BID, NOXA, and BAX in an Ehrlich tumor model. Curcumin, from *Curcuma longa*, reduced tumor multiplicity and colonic inflammation in an AOM-DSS-induced CRC model. These studies suggest shared anti-cancer mechanisms such as apoptosis induction and inflammation suppression, despite differences in plant origin and animal model types (created by the authors using PowerPoint).

**Figure 4 curroncol-32-00546-f004:**
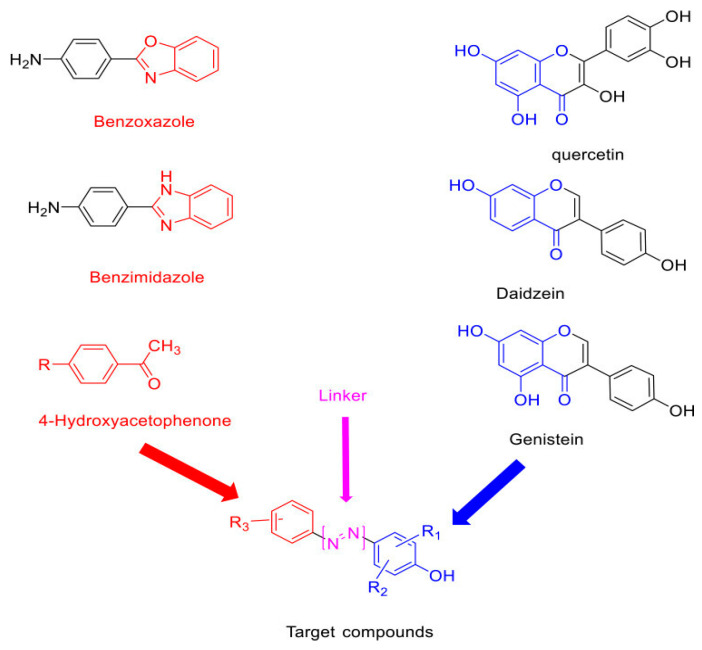
Structural components and molecular hybridization strategy of dual-target CRC inhibitors. Compounds C4 and G4 are developed via scaffold fusion between synthetic benzoxazole/benzimidazole backbones and natural flavonoids (quercetin, daidzein, genistein), with 4-hydroxyacetophenone used as a reactive linker. This design enables simultaneous inhibition of EGFR and COX-2 in colorectal cancer cells. Redrawn and adapted from [30] Abdelgawad MA, et al. Drug Des Devel Ther. 2021; 15:3871–3891. https://doi.org/10.2147/DDDT.S310820. Licensed under CC BY 4.0.

**Figure 5 curroncol-32-00546-f005:**
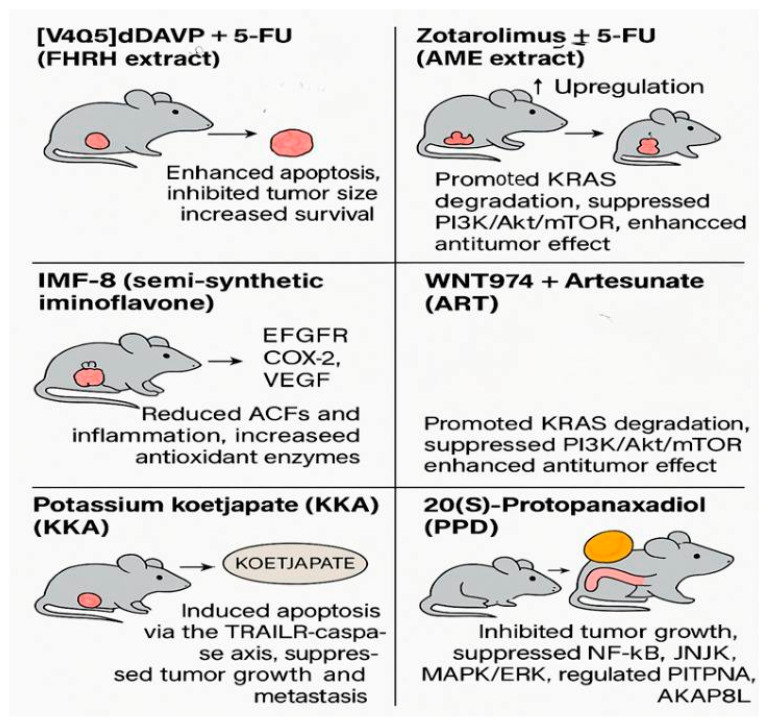
Graphical summary of synthetic compounds evaluated in vivo for colorectal cancer treatment. The figure illustrates tumor suppression, apoptosis induction, and molecular pathway modulation in animal models following treatment with various agents, including [V4Q5]dDAVP + 5-FU, Zotarolimus ± 5-FU, IMF-8, WNT974 + ART, Potassium Koetjapate (KKA), and 20(S)-Protopanaxadiol (PPD) (adapted from Table 4 data, based on studies [39,40,41,42,43,44]) (created by the authors using PowerPoint).

**Figure 6 curroncol-32-00546-f006:**
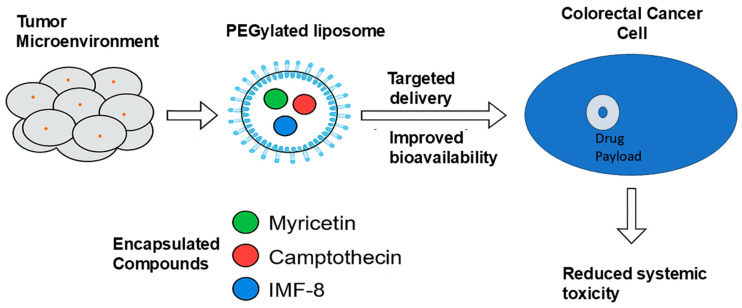
Schematic illustration of PEGylated liposomes for targeted delivery of anti-colorectal cancer agents. Compounds such as myricetin, camptothecin, and IMF-8 are encapsulated and delivered via PEGylated liposomes to the colorectal tumor site, improving bioavailability and reducing systemic toxicity (created by the authors using PowerPoint).

**Figure 7 curroncol-32-00546-f007:**
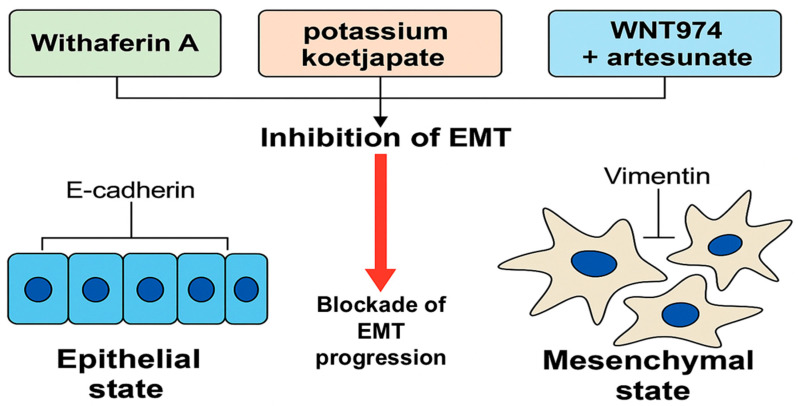
Inhibition of epithelial–mesenchymal transition (EMT) in colorectal cancer (CRC) by selected natural and semi-synthetic compounds. EMT is characterized by the downregulation of epithelial markers such as E-cadherin and the upregulation of mesenchymal markers like vimentin, facilitating cancer cell motility and invasiveness. Natural agents such as Withaferin A, potassium koetjapate (KKA), and the WNT974–artesunate combination have shown preclinical potential to block EMT progression in CRC models. EMT modulation represents a promising therapeutic strategy to inhibit metastasis and improve prognosis in CRC (created by the authors using PowerPoint).

**Table 1 curroncol-32-00546-t001:** In vitro studies of natural products against colorectal cancer cell lines, highlighting mechanisms of action and target pathways.

Natural Product	Study Type	Target Cell Line(s)	Key Mechanism/Findings	Author
*Nigella sativa* (CSENS extract)	In vitro	HCT116	Induced apoptosis via NF-κB, AP-1, Nrf2 modulation; Bax/Bcl-2 regulation	Ayman I Elkady, 2015 [7].
Olive Leaf Extract (chlorogenic acid)	In vitro	HT29	Induced S-phase arrest, ROS generation, apoptosis	Albogami, S., 2021 [18].
*Ferula hermonis* (FHRH extract)	In vitro	LoVo	Caspase 3/7 activation; apoptosis; bioactive compounds identified (Alpha-Bisabolol, Baccatin III)	Abutaha, N., Nasr; 2019 [19].
Flavonoids (general)	In vitro	Various colon cancer lines	Promoted apoptosis, suppressed proliferation via multiple pathways	Mariam Abotaleb, 2018 [8].
AgNPs (*Lasiurus scindicus* and *Panicum turgidum*)	In vitro	HCT116	Green-synthesized nanoparticles showed cytotoxicity against cancer cells	Alburae, N., 2024 [20].
*Selaginella repanda* (ethanolic extract)	In vitro	HCT116	Induced apoptosis, dose/time-dependent cytotoxicity	Adnan, M., 2021 [21].
*Sansevieria trifasciata* (ethanolic extract)	In vitro	HCT116	Selective cytotoxicity toward cancer cells, reduced IC_50_ vs. normal colon cells	Afzal, S., 2024 [22].
*Tetraclinis articulata* (essential oil)	In vitro	SW620	Moderate cytotoxicity (IC_50_ < 30 μg/mL), attributed to oxygenated sesquiterpenes (caryophyllene oxide, carotol)	Jlizi S, 2021 [23].
*Moringa oleifera* (leaf and bark extracts)	In vitro	HCT-8	Induced apoptosis, G2/M phase arrest, bioactive compounds (eugenol, D-allose)	Abdulrahman Khazim Al-Asmari, 2015 [14].
*Ziziphus nummularia* (ethanolic extract)	In vitro	HCT8	Apoptosis induction; microtubule disruption via luteolin-7-O-glucoside	Alghamdi, S.S., 2024 [24].
*Rhazya stricta* (CAERS extract)	In vitro	HCT116	Downregulated NF-κB/AP-1; upregulated p53, caspase-3/7/9, Bax	Elkady, A.I., 2016 [25].

**Table 2 curroncol-32-00546-t002:** In vivo studies on anti-tumor effects of natural products in colorectal cancer animal models.

Natural Products	Study Type	Animal Model	Key Mechanism/Findings	Author
*Ferula hermonis* (FHRH extract)	In vivo	DMBA-induced mammary tumor model (rodent)	Induced apoptosis, reduced tumor size, identified bioactive compounds (Alpha-Bisabolol, Baccatin III)	Abutaha, N., Nasr; 2019 [19].
*Ferula assa-foetida* (OGR extract)	In vivo	HT-29 xenograft mouse model	Reduced tumor volume and induced apoptosis via PUMA, BIM, BIK, BAK upregulation	Elarabany, N., 2023 [27].
*Arthrocnemum machrostachyum* (AME extract)	In vivo	Ehrlich solid tumor (EST) model in mice	Induced apoptosis, reduced tumor size, and modulated apoptotic gene expression	Sharawi, Z.W., 2020 [28].
Curcumin (*Curcuma longa*)	In vivo	AOM-DSS-induced CRC in high-protein diet-fed mice	Reduced tumor multiplicity, decreased inflammation, colonocyte proliferation, and toxic metabolites	Byun, S.-Y., 2015 [29].

**Table 3 curroncol-32-00546-t003:** In vitro studies of synthetic and semi-synthetic compounds in colorectal cancer treatment, with target cell lines and key outcomes.

Compound	Study Type	Target Cell Lines (s)	Key Findings	Author
C4, G4 (semi-synthetic derivatives)	In vitro	HT-29 (CRC), PaCa-2, A375, H-460, Panc-1	Dual EGFR and COX-2 inhibition; potent cytotoxicity	Abdelgawad, M.A., 2021 [30].
Silver nanoparticles using *Chamomile* (SN-CHM)	In vitro	SW620, HT-29 (CRC)	Induced apoptosis via BAX upregulation; reduced cell viability	Abdellatif, A. a. H., 2023 [31].
Withaferin-A + 5-FU (combination therapy)	In vitro	CRC cell lines (specific lines not detailed)	Induced ER stress-mediated apoptosis and autophagy; β-catenin inhibition; G2/M arrest	Alnuqaydan, A.M., 2020 [32].
Myricetin-conjugated silver nanoparticles	In vitro and in silico	CRC cell lines	Induced apoptosis; confirmed cytotoxicity; supported by TCGA analysis	Anwer, S.T., 2022 [33].
Polar extract of *Adansonia digitata* fibers	In vitro	HCT116 (CRC), MCF-7	Inhibited proliferation; modulated gene expression including CSNK2A3 and FGD3	El-Masry, O.S., 2021 [34].
Cobalt oxide nanoparticles (green synthesis from *Psidium guajava*)	In vitro	HCT116 (CRC), MCF-7	Reduced cancer cell viability; antibacterial and photocatalytic properties	Govindasamy, R., 2022 [35].
Oxazole analogues (compound 14)	In vitro	HCT116 (CRC)	Antiproliferative activity (IC_50_ = 71.8 μM); strong CDK8 binding	Shaikh MAS., 2025 [36].
Camptothecin–CEF nanocomposite	In vitro	HT29 (CRC), A549	Improved CPT delivery; induced apoptosis via caspase-3; G1 phase arrest	Krishnan, P., 2017 [37].
Sipholenol A-4-O-acetate, Sipholenol A-4-O-isonicotinate	In vitro	P-gp-overexpressing cancer cell lines	Reversed MDR by inhibiting P-gp efflux; increased paclitaxel retention	Zhang, Y., 2015 [38].

**Table 4 curroncol-32-00546-t004:** In vivo evaluation of synthetic compounds in animal models for colorectal cancer, including tumor inhibition and molecular effects.

Compound	Study Type	Animal Model	Key Findings	Author
[V4Q5]dDAVP + 5-FU	In vivo	CT-26 and COLO-205 tumor-bearing mice	Enhanced 5-FU efficacy; inhibited tumor growth and metastasis; increased survival	Sobol, N.T., 2023 [39].
Zotarolimus ± 5-FU	In vivo	HCT-116 xenograft in BALB/c nude mice	Reduced tumor growth; enhanced apoptosis; downregulated EGFR, COX-2, VEGF	Chang, G.-R., 2021 [40].
IMF-8 (semi-synthetic iminoflavone)	In vivo	DMH-induced CRC in rats	Reduced ACFs and inflammation; increased antioxidant enzymes	Prasad, V.G., 2014 [41].
WNT974 + Artesunate (ART)	In vivo	CRC xenograft mouse model	Promoted KRAS degradation; suppressed PI3K/Akt/mTOR; enhanced antitumor effect	Gong, R.-H., 2022 [42].
Potassium koetjapate (KKA)	In vivo	HCT116 xenograft in nude mice	Induced apoptosis via the TRAILR-caspase axis; suppressed tumor growth and metastasis	Jafari, S.F., 2024 [43].
20(S)-Protopanaxadiol (PPD)	In vivo	HCT116 xenograft in nude mice	Inhibited tumor growth; suppressed NF-κB, JNK, MAPK/ERK; regulated PITPNA, AKAP8L	Gao, J.-L., 2013 [44].

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
