# Peer review of "Natural and Synthetic Compounds Against Colorectal Cancer: An Update of Preclinical Studies in Saudi Arabia"

_curroncol, 2025, doi:10.3390/curroncol32100546_

Round 1
Reviewer 1 Report (Previous Reviewer 2)
Comments and Suggestions for Authors
This novel version of the manuscript presents some improvements. The overall message remains excessively broad and unfocused. I still believe that the manuscript is too long. But this analysis may be published to illustrate the modest contribution of Saudi Arabia to cancer research.
Corrections needed:
- Chemical formula. The chemical formula shown in Fig 3 (top left panel) is incorrect and must be redrawn. What is this bizarre compound? Fig 4. The structure of 4-hydroxyacetophenone is not correct (R = OH). Formula in Fig 5 are also erroneous.
- The Graphical Abstract contains too much text. A GA should be essentially “graphical” to catch the attention of readers. The sun-like model in the center is useless.
Author Response
Please see the attachment.

Reviewer 2 Report (New Reviewer)
Comments and Suggestions for Authors
Comments
The manuscript with entitled “ Natural and Synthetic Compounds Against Colorectal Cancer: An Update of Preclinical Studies in Saudi Arabia”.
- This manuscript discusses the protective role of natural and synthetic compounds against colorectal cancer. This topic is highly relevant and interest in health. It is intriguing. The manuscript is original.
- In the introduction, The indicators of anti-cancer effect should be introduced. Recommendation: The author to provide additional clarification.
- In the methodology, the article lacks writing methods for this review, such as data and literature sources. The author should provide this section.
- In the results, in many paragraphs and sections, there is only one reference, and more references should be added. In the introduction of anti-cancer extracts in the article, there is no mention of the main compounds in these extracts. These compounds should be introduced.
- In the conclusions, the limitations of the research and the introduction of future research directions are well presented..
7. The format of references is not consistent.
Comments on the Quality of English Language- There are some grammar and style errors, especially, the capitalization of words.
- The Latin name format of the species should meet the requirements.
Round 2
Reviewer 1 Report (Previous Reviewer 2)
Comments and Suggestions for Authors
My previous points have been considered, at least in part. No other comments.
Reviewer 2 Report (New Reviewer)
Comments and Suggestions for Authors
None
This manuscript is a resubmission of an earlier submission. The following is a list of the peer review reports and author responses from that submission.
Round 1
Reviewer 1 Report
Comments and Suggestions for Authors
The search for new therapeutic possibilities for an increasingly large population of patients affected by colorectal cancer is certainly a clinical priority. However, in light of current knowledge, the data on the effectiveness of both synthetic and natural compounds remain very weak.
- The content of the proposed review mostly consists of an almost sterile list of natural and synthetic compounds, which at times becomes repetitive. Given the aim of the review, and also to make the text more engaging, the authors should adopt a more critical approach, which would also confer greater significance to the work.
- In some sections, the objective of the review appears somewhat confused. Certain compounds, such as curcumin, are described as “dietary chemopreventive agents, particularly in populations with high-risk dietary patterns,” which distances them from the concept of therapeutic strategies for CRC as stated in the title.
- Overall, the review lacks a clear narrative structure or cohesive framework that would help the reader understand the mechanistic significance of each compound discussed. It presents a compilation of data from various studies without critically analyzing discrepancies between results or highlighting areas that warrant further investigation. As a result, the manuscript lacks the depth expected from a scholarly review and remains at the level of a descriptive summary.
Author Response
Authors comments are attached.

Reviewer 2 Report
Comments and Suggestions for Authors
This atypical review aimed at analyzing the treatment strategies for colorectal cancer (CRC) in Saudi Arabia. In fact, the review is essentially a collection of CRC laboratory studies, not clinical studies, performed in Saudi Arabia. The title is misleading and should be completely revised. Something like “Natural extracts, natural substances and synthetic compounds active against colorectal cancer: an update of studies in Saudi Arabia” would be more appropriate. In fact, I don’t understand why it is useful to focus the study on a single country? A better justification should be provided.
Section 1. Study aims. is redundant with the end of the Introduction (section 2). It can be removed. Section 2 contains information about specific products and plants, redundant with information in section 3. The organization of the manuscript should be reworked.
Importantly, the review is not illustrated. There is not a single Figure and this is unacceptable. When non-classical compounds are cited, their chemical structures should be given. For ex, the citation of phenolic acid derivatives C4-G4 or flavin IMF-8 presents no-interest if the corresponding structures are not shown. At least 2-3 Figures should be included to illustrate specific compounds and their mechanism of action.
Section 5 Clinical application refers essentially to non-clinical studies with extracts and experimental products, not clinical studies. Here again, the organization is not coherent.
The last sections 6-7-8 are quite verbose, mainly rehashing familiar concepts and ideas. Lack of innovative ideas and proposals. The three sections may be combined and new ideas introduced.
References in the Tables are not numbered.
The literature survey is not complete. The incidence and treatment of CRC in Saudi Arabia should be better discussed. There are multiple CRC studies focused on Saudi Arabia to cite.
In brief, the present review is relatively weak and lacks innovation. The most important point is to better justify the focus on CRC studies in Saudi Arabia only. CRC is a general health problem, there is no reason to restrict the analysis to a single country, especially for a discussion centered on experimental products (non-clinical).
Author Response
Authors comments are attached.

Round 2
Reviewer 2 Report
Comments and Suggestions for Authors
Some efforts have been made to improve the manuscript, notably to better justify the topic. However, the manuscript remains relatively weak and superficial, lacking a strong new message. If it is published… the figures must be reworked:
Fig 1 is totally wrong. Sipholenol indicated twice. G2/M Typo errors (checkpint/checkpont). Cycle phases wrongly indicated (S/S), etc.
Fig 4 with wrong structure/name (R, 4-hydroxyacetophe). Compound C4-G4 largely cited (even in the abstract) should be fully presented, otherwise the text is incomprehensible.
Fig 5: typing errors (promoted, extra¨ct).
Fig 6 very poor symbolism. Redraw.
What is the value of Fig 7? The effects of the studied
Author Response
Comment 1:
Fig 1 is totally wrong. Sipholenol indicated twice. G2/M Typo errors (checkpint/checkpont). Cycle phases wrongly indicated (S/S), etc.
Response 1:
Thank you for pointing this out. We agree with the concern. Figure 1 has been completely revised to correctly indicate the cell cycle checkpoints (G1, S, G2/M) and to map which natural compound affects each phase. The updated figure reflects more accurate cell cycle stages and includes mechanisms of apoptosis, as now shown for compounds such as Moringa oleifera (G2/M arrest), Olea europaea (S-phase arrest), and Ferula hermonis (apoptosis).
[Updated Figure 1, Page 6 of 23]
Comment 2:
Fig 4 with wrong structure/name (R, 4-hydroxyacetophe). Compound C4-G4 largely cited (even in the abstract) should be fully presented, otherwise the text is incomprehensible.
Response 2:
Thank you for highlighting this important issue. We agree with the reviewer’s observation.
- The structural depiction of 4-hydroxyacetophenone in Figure 4 has now been corrected, and the compound’s full chemical name is clearly labeled.
- We also revised the figure caption to clarify the scaffold hybridization strategy used in C4–G4 compound design (fusion of benzoxazole, benzimidazole, and flavonoid pharmacophores).
- Additionally, a new paragraph has been added in the Results section (Page 10 of 23, Paragraph 345-355 line) that explicitly explains the molecular structure, rationale, and dual-target mechanism of compounds C4 and G4, including their ICâ‚…â‚€ values, synthetic origin, and interaction with EGFR and COX-2 pathways. This helps contextualize their significance, especially since they are referenced in the abstract and conclusion.
We appreciate your suggestion, which has substantially improved the manuscript’s clarity and consistency.
[Updated Figure 4, Page 10 of 23]
Comment 3:
Fig 5: typing errors (promoted, extra¨ct).
Response 3:
Thank you for pointing out the typographical errors in Figure 5. We have carefully reviewed the figure and corrected the identified issues.
[Updated Figure 5, Page 16 of 23]
Comment 4:
Fig 6 very poor symbolism. Redraw.
Response 4:
Thank you for this valuable feedback. We acknowledge that the original version of Figure 6 lacked visual clarity and symbolic precision. In response to your suggestion:
- Figure 6 has been completely redrawn with improved visual elements to enhance readability, clarity, and scientific symbolism.
- The updated schematic now clearly illustrates the PEGylated liposomes for targeted delivery of anti-colorectal cancer agents, showing how natural and synthetic agents (e.g., myricetin, camptothecin, IMF-8) are encapsulated in PEGylated liposomes.
- We added better-labeled compartments, directional arrows indicating drug release and tumor targeting, and standardized icons for molecules, liposomes, and tumor cells to align with MDPI visual standards.
- The caption of Figure 6 has also been revised to accurately reflect the improved figure contents and highlight the translational relevance of the delivery system.
We believe these improvements address the reviewer’s concern and result in a figure that is more informative and visually engaging for readers.
[Updated Figure 6, Page 17 of 23]
Comment 5:
What is the value of Fig 7? The effects of the studied
Response 5:
We thank the reviewer for this important observation. In response, Figure 7 has been revised to directly reflect the preclinical findings highlighted in our manuscript. Specifically, it now illustrates the inhibition of epithelial–mesenchymal transition (EMT) in colorectal cancer (CRC) by natural and semi-synthetic compounds investigated in Saudi-based studies, including Withaferin A, potassium koetjapate (KKA), and the combination of WNT974 and artesunate (ART).
These agents were shown in preclinical models to suppress EMT-related processes such as E-cadherin downregulation and vimentin upregulation, thereby reducing metastatic potential and invasiveness. EMT is a key contributor to CRC progression, metastasis, and therapy resistance. Thus, this figure provides a mechanistic summary linking the reviewed compounds to EMT modulation, supporting their potential relevance in translational oncology.
We believe this updated figure now better integrates with the article’s content and enhances the mechanistic clarity of the discussed agents.
[Updated Figure 7, Page 18 of 23]
please see the attachment

Round 3
Reviewer 2 Report
Comments and Suggestions for Authors
Some improvements